# Assessment of Psychological Burden in Individuals with Hereditary Risk of Pancreatic Cancer Under Surveillance: Evaluation of Distress 3 Years After Enrollment

**DOI:** 10.3390/cancers17183014

**Published:** 2025-09-16

**Authors:** Veronica Marinelli, Maria Angela Mazzi, Olga Maggioni, Elisa Venturini, Michela Rimondini, Michele Milella, Salvatore Paiella, Roberto Salvia

**Affiliations:** 1Department of Engineering for Innovative Medicine, University of Verona, 37134 Verona, Italy; elisa.venturini@univr.it (E.V.); michele.milella@univr.it (M.M.); roberto.salvia@univr.it (R.S.); 2Pancreas Institute, University of Verona Hospital Trust, 37134 Verona, Italy; 3Department of Neurosciences, Biomedicine and Movement Sciences, University of Verona, 37134 Verona, Italy; mariangela.mazzi@univr.it (M.A.M.); olga.maggioni@univr.it (O.M.); michela.rimondini@univr.it (M.R.); 4Department of Surgery, Dentistry, Paediatrics and Gynaecology, University of Verona, 37134 Verona, Italy

**Keywords:** pancreatic cancer, hereditary risk, surveillance, psychological burden, distress

## Abstract

Pancreatic cancer (PC) is one of the most lethal cancers, with early detection being critical for improving long-term survival through surgery and chemotherapy. Late-stage diagnosis significantly worsens prognosis, prompting increased surveillance for high-risk individuals (HRI) to enable earlier detection. However, there is a lack of longitudinal studies on the emotional and psychological impacts of PC surveillance. This study evaluates the distress experienced by 44 HRI participants over three years in the Italian Registry of Families at Risk of Pancreatic Cancer, receiving annual MRI scans and consultations at the Verona Pancreas Institute. The current longitudinal study provides valuable insights into the evolving psychological experiences of high-risk individuals (HRIs) for PC over a 3-year period. By examining key variables such as perceived stress, self-efficacy, coping strategies, and social support, the research deepens our understanding of the psychosocial challenges faced by this vulnerable group. These findings can contribute to the development of targeted interventions aimed at improving mental health and overall well-being among HRIs, thereby supporting the adoption of more effective coping mechanisms and enhancing quality of life during this high-risk period.

## 1. Introduction

Pancreatic cancer (PC) is a clinically challenging cancer, due to both its late stage at diagnosis and its resistance to chemotherapy [1].

Early PC (stage I) is rarely detected in clinical practice [2]. Surgical resection with adjuvant systemic chemotherapy currently provides the only chance of long-term survival [3]; however, only 15–20% of patients are eligible for surgery at diagnosis [4] because the majority are diagnosed at a late stage [5,6].

Over the past decades, multiple surveillance programmes and research consortia have been developed worldwide to actively monitor enriched cohorts of HRIs [7,8,9,10], aiming to anticipate PC diagnosis at a preclinical stage.

Significant progress has been made in pancreatic cancer surveillance programmes targeting high-risk populations; recent evidence suggests that PC surveillance is both feasible and safe, also within a fully public health care system [11], resulting in early diagnoses of PC or premalignant lesions, primarily at baseline but also over time [12,13,14].

Despite these successes, little is known about how surveillance impacts the emotional and psychological well-being of HRIs. It remains unclear whether surveillance ultimately improves quality of life by reducing cancer-related worries. Similar psychological impacts have been studied extensively in breast, ovarian, and colon cancer screening programmes [15,16,17] with large cohorts.

To date, only a few studies have examined the psychological feasibility of PC surveillance, all reporting a consistently low psychological burden among participants [18,19,20].

A recent systematic review of seven studies concluded that HRIs might benefit from enrollment in PC surveillance, exhibiting reduced anxiety over time and experiencing low to moderate PC-related distress [21].

Nonetheless, it is reasonable to question whether prolonged participation in PC surveillance programmes imposes any psychological burden over time on HRIs with heredo-familial predisposition.

The study aimed to evaluate the distress experienced by HRIs 3 years after engaging in PC screening at Pancreas Institute of Verona and to describe which participant characteristics contributed to the increase in stress.

## 2. Methods

### 2.1. Study Design and Data Collection

This is a longitudinal follow-up of a previously published cohort [22].

The subjects enrolled at baseline in this study were recruited from the Verona Pancreas Institute. The study received approval from the local ethics committee (#462CESC). Participants belonged to the Italian Registry of Families at Risk of Pancreatic Cancer (IRFARPC) [11,23]. Most participants were relatives of patients previously treated for PC at the General and Pancreatic Unit of the Pancreas Institute, often enrolled during their relative’s hospitalisation.

Inclusion criteria included: having three or more relatives affected by PC within up to the third degree of kinship, or two relatives affected if at least one is a first-degree relative; possessing a known genetic mutation in BRCA1, BRCA2, ATM, PALB2, or CDKN2A genes with at least one affected first- or second-degree relative; or having a previous diagnosis of hereditary pancreatitis or Peutz-Jeghers syndrome. According to the surveillance protocol, each enrolled individual undergoes an annual baseline magnetic resonance cholangiopancreatography (MRCP).

For patients who decline or are unable to undergo MRCP, endoscopic ultrasound is offered as an alternative. In addition to these medical diagnostic procedures, for this longitudinal study starting in July 2017, participants have also been offered an individual meeting with a clinical psychologist. The initial screening of the emotional and psychological impact of the surveillance programme has already been published [22]. All enrolled participants provided written informed consent.

### 2.2. Procedures

HRIs with heredo-familial predisposition undergoing PC surveillance with MRCP and a clinical visit were re-evaluated in the same psychological assessment (perceived stress, self-efficacy, coping abilities, and social support) three years after the baseline assessment, from 2019 to 2023. The follow-up assessments were conducted by a clinical psychologist specialising in psycho-oncology via telephone after the radiological and clinical evaluations were made. The subjects were contacted by phone for a follow-up screening 36 months after their initial psychological assessment. The average duration of the phone interview was 30 min. HRIs were contacted up to three times; if there was no response after these attempts, they were considered lost to follow-up.

### 2.3. Questionnaires

The following assessment questionnaires were used to investigate different functioning areas, as summarised in Table 1.

#### 2.3.1. Perceived Stress Scale (PSS)

PSS, created by Cohen et al. [24,25], measures stress as it is perceived, looking into feelings and thoughts experienced by the subject in the previous month. The questionnaire is designed to assess at what stress level the subject feels his life is burdensome and unbearable. A global score and two subscales, which inform on the Perceived Helplessness (items 1, 2, 3, 6, 9,10) and Lack of Self-Efficacy (items 4, 5, 7, 8), can be obtained by summing the response scores. A set of cut-offs is used to interpret the global score, although the scale is not a diagnostic tool: 13 or lower is considered normal, from 14 to 26 moderate, and 27 or higher is severe stress [26]. The PSS-10 was translated into Italian by Andrea Fossati [27]; a recent validation study [28] confirmed the bi-factor structure also for the Italian version. In this study, the Cronbach’s α was 0.86 for the global score, 0.86 for Perceived Helplessness and 0.74 for Perceived Self-Efficacy.

#### 2.3.2. General Self-Efficacy Scale (GSES)

Perceived self-efficacy is the belief that one can perform a difficult task or cope with adversity. It relates to problem-solving abilities that the subject would use to adapt after experiencing stressful life events. It can be referred to as a positive resistance resource. GSES [29] is correlated to positive emotions, optimism and work satisfaction; negative correlations were found for depression, stress, health complaints, burnout and anxiety. The final score is between 10 and 40, where higher scores relate to higher self-efficacy levels. An Italian version of the scale was used in this study [30], which showed a good reliability (Cronbach’s α = 0.90).

#### 2.3.3. Coping Orientation to Problems Experienced (Brief COPE)

This is a short version [31] of the COPE questionnaire [32,33]. The test evaluates the subject’s coping style through investigation of problem-solving abilities and emotional fluctuation, as a response to stressful situations. It explores 14 different coping strategies, with the use of two different items each. Every item can score between 1 (“I haven’t been doing this at all”) and 4 (“I have been doing this a lot”). The overall score reflects the extent of coping strategy use, with higher values indicating greater reliance, which may be either adaptive or maladaptive. The 14 subscales provide more detailed, coping-specific information, each ranging from 2 to 8. A higher-order classification was suggested by Cooper and colleagues [34], which identified a functional model based on three different coping styles: emotion-focused (range: 10–40), problem-focused (range: 6–24), and dysfunctional strategies (range: 12–48). The reliability of this study was 0.90 for the global score, 0.75 for the problem-focused strategies, 0.70 for the emotion-focused and 0.53 (here, the items 4 and 11, concerned with substance use, were not informative, due to constant responses) for the dysfunctional strategies scale.

#### 2.3.4. The Multidimensional Scale of Perceived Social Support (MSPSS)

MSPSS [35] is designed to explore the subject’s social support system. It measures the perceived support from three different sources: family (items 3, 4, 8, 11), friends (items 6, 7, 9, 12) and a significant other (1, 2, 5, 10). The scale consists of 12 items, with 4 items for each support area. Each item is rated on a 6-point Likert-type scale, ranging from 1 (“very strongly disagree”) to 6 (“very strongly agree”); higher scores indicate greater perceived social support, both in the global scale and specific area scales [36]. An Italian version of the scale by Prezza and Principato [37] was used for this study (α = 0.98 for the global score, 0.97 for the friend support and α = 0.99 for the family and other significant person scales).

### 2.4. Statistical Analysis

Quantitative variables were reported as means and standard deviations, while qualitative variables were summarised using frequencies and percentages. Normality of data distribution was assessed through skewness and kurtosis tests. For comparisons between baseline and follow-up values, the paired *t*-test was applied when normality assumptions were met; otherwise, particularly in the exploration at the item level, the non-parametric Wilcoxon signed-rank test was used. Given the small group sizes, comparisons across the three independent levels of psychological distress (PSS classes) were conducted using the Kruskal–Wallis test. A *p*-value less than 0.05 was considered statistically significant. All analyses were performed using Stata 18 (StataCorp. 2023. Stata Statistical Software: Release 18. College Station, TX, USA: StataCorp LLC).

## 3. Results

Of the 54 HRIs initially evaluated at baseline, ten did not respond to phone contacts and have been classified as lost to follow-up. The remaining 44 HRIs, of whom 29 (65.9%) were female, with a mean age of 56.1 years (SD = 10.2), agreed to participate in a follow-up psychological assessment conducted three years after their initial evaluation. Among these participants, 20 were aged between 18 and 64 at the time of follow-up, while 24 were 65 years or older. Additionally, seven participants (15.9%) experienced the loss of a family member to PC during the three-year study period.

### 3.1. Outcomes of the Distress Assessment Conducted After 36 Months

#### 3.1.1. Perceived Stress Scale (PSS)

Our sample scored a mean of 16.9 (SD = 7.7) in the PSS questionnaire, which means that compared to three years ago (mean = 14.8, SD = 6.1; Table 2), the perceived stress of these subjects tends to have deteriorated, in both domains. Figure 1 shows individual data points and evidences the suggested cut-scores, so that 68% of the responders now objectively experience psychological distress (59% moderate and 9% severe levels), while at baseline, they were 55% (50% moderate and 5% high values). A more detailed comparison, at the item level (Figure 2), highlights that the same trend of variation involves both the items of the Helplessness scale, which are negatively worded items, and those of the Lack of Self-efficacy scale (positively worded); the changes between the baseline and follow-up evaluations are statistically relevant for items 8 and 9 (Wilcoxon z = −2.34, *p* = 0.02 for both).

#### 3.1.2. General Self-Efficacy Scale (GSES)

A decrease in own problem-solving skills, as gathered by the GSES questionnaire, is observed in the subjects of our sample (Baseline: 32.02; Follow-up: 28.09, *p* < 0.01; see Table 2). Three-quarters of respondents (33/44) shifted down their beliefs about the ability to control events, with a remarkable change of about 6 points, corresponding to a 20% scale-wide (mean value 5.6, SD = 3.3; range: 1–14).

#### 3.1.3. Coping Orientation to Problems Experienced (Brief COPE)

The global score of the Brief-COPE questionnaire demonstrates time instability in coping abilities (Baseline: 64; Follow-up: 61, *p* = 0.05), primarily due to the problem-solving area, which shows a relevant time change (from 18.80 to 17.45, *p* < 0.01). What emerges from a more detailed analysis, comparing the 14 specific strategies in sequential observations (Table 3), confirms relevant changes in “planning” (Δ = 0.68), which may indicate a decrease in perceived operational control over problems. Conversely, within the set of dysfunctional coping strategies, a significant reduction in the use of “denial” can be observed (Δ = 0.52), suggesting an increased awareness of reality.

#### 3.1.4. The Multidimensional Scale of Perceived Social Support (MSPSS)

The MSPSS scoring frequency at the follow-up was slightly under baseline average (4.27 vs. 4.59, *p* = 0.05), still showing the perception of a good overall social support. The subscale values show that this reduction was attributable to the friend and other relevant person areas, which show a difference of about half a point on the Likert scale (Δ = 0.55 and 0.45, respectively); while the family area gave constant support (Δ = 0.08).

### 3.2. Outcomes of the Psychological Evaluation Conducted After 36 Months

Based on the demographic and psychological characteristics of the 44 study participants, stratified according to stress levels reassessed at follow-up, three distinct clusters were identified:Normal Stress: individuals with a PSS score below 14;Stable Clinical Stress: individuals with elevated stress both at baseline and at follow-up (PSS at follow-up >14 and PSS at baseline >14);New Clinical Stress: individuals who developed elevated stress at follow-up for the first time (PSS at follow-up >14 and PSS at baseline <14).

Results indicate that, after 36 months from the initial assessment, fourteen individuals (32% of the sample) in the Normal Stress group, with a mean age of 54.6 years and predominantly female (64%), showed no stress scores comparable to baseline. Among them, one individual experienced the loss of a first-degree relative to pancreatic cancer during the three-year follow-up.

This profile highlights individuals with a high level of self-efficacy, reflecting stable confidence in their ability to handle daily challenges. The average score on the General Self-Efficacy Scale was 32.2 (SD: 5.8), indicating a positive perception of their competence and resilience. Furthermore, their overall adaptation appears adequate, as evidenced by a score of 61.1 (SD: 8.8) on the Brief COPE, suggesting the use of effective stress management strategies and avoidance of maladaptive behaviours. The low frequency of maladaptive coping strategies is reinforced by a score of 20.4 (SD: 2.8) on the COPE-dysfunctional subscale. The preferred coping mechanisms are problem-solving-oriented (COPE-problems FU score: 17.6, SD: 2.9) and emotion regulation strategies (COPE-emotions FU score: 23.20, SD: 4.6), both considered more effective for managing stress compared to dysfunctional behaviours. Regarding perceived social support, individuals rated their support network positively, with a total MSPSS score of 4.7 (SD: 0.9). Notably, support from family was perceived as particularly strong (MSPSS family FU score: 5.3, SD: 1.0), while support from friends and significant others was also of good quality (MSPSS friends FU score: 4.1, SD: 1.0; MSPSS other persons FU score: 4.8, SD: 1.0).

The sample comprised 20 individuals experiencing Stable Clinical Stress (46% of the total sample), with a mean age of 56.3 years and predominantly female (60%). These individuals exhibited moderate to severe stress levels comparable to baseline. During the three-year follow-up, six participants (30%) reported losing a first-degree relative to pancreatic cancer.

This profile corresponds to individuals who scored an average of 25.8 (SD = 5.4) on the General Self-Efficacy Scale (GSES), reflecting a moderate level of confidence in their ability to manage challenging situations. Their overall adaptation to difficulties, as measured by the Brief COPE, was adequate, with a total score of 58.4 (SD = 10.0), indicating the use of generally functional coping strategies.

On the COPE-emotional subscale, participants scored an average of 20.7 (SD = 6.3), suggesting a tendency to employ emotion-focused coping strategies such as seeking emotional support and reflecting on their feelings. The Multidimensional Scale of Perceived Social Support (MSPSS) yielded a total score of 3.6 (SD = 1.3), indicating moderate perceived social support. Notably, support from friends was relatively limited, with a score of 3.3 (SD = 1.3).

Ten individuals (22% of the sample) in the New Clinical Stress group, with a mean age of 59.9 years and predominantly female (80%), exhibited moderate to severe stress levels of new onset. None of these individuals reported losing a first-degree relative to pancreatic cancer during the three-year follow-up period.

This profile corresponds to individuals who scored 27.0 on the General Self-Efficacy Scale (GSES), with a standard deviation of 3.4, reflecting a moderate level of confidence in their ability to manage various situations. Their total score of 67.7 (SD 7.2) on the Brief COPE indicates an excellent and effective capacity for coping with difficulties, characterised by generally functional stress management strategies and a preference for emotion-focused approaches (mean 25.3, SD 5.7). Additionally, they scored 5.0 (SD 0.4) on the Multidimensional Scale of Perceived Social Support (MSPSS), reflecting very good perceived social support. Notably, support from family and significant others was particularly prominent, with scores of 5.5 (SD 0.5) and 5.2 (SD 0.4), respectively.

It is important to emphasise that, across all three groups, regardless of the perceived stress level, individuals tend to adopt dysfunctional coping strategies to only a minimal extent (see Table 4). It is also interesting to note the distribution of mean changes in the PSS-10 items at times of engagement and after 36 months: the subjects, following the follow-up, exhibit a decrease in self-efficacy and an increase in feelings of helplessness (Figure 2).

## 4. Discussion

This longitudinal study offers valuable insights into the evolving psychological landscape of 44 HRIs for PC over a three-year period. By examining variables such as perceived stress, self-efficacy, coping strategies, and social support, the findings provide a nuanced understanding of the psychosocial challenges faced by this vulnerable population.

Regarding perceived stress, participants demonstrated an increase in scores on the PSS, rising from an average of 14.8 to 16.9. This worsening can be attributed to several factors: awareness of their risk status, anxiety related to awaiting potential diagnoses, and the possible loss of a family member due to pancreatic cancer.

The observed increase in perceived stress aligns with prior research [38], indicating that individuals at high genetic or familial risk for cancer often experience escalating anxiety and distress over time. The rise from an average PSS with 68% now experiencing moderate or severe distress, underscores the persistent psychological burden associated with ongoing risk awareness and surveillance procedures. This supports the working hypothesis that prolonged exposure to health-related uncertainty exacerbates stress levels, potentially impairing quality of life.

Additionally, analysis of individual items on the PSS indicated that feelings of helplessness and lack of self-efficacy are particularly sensitive to temporal changes, with statistically significant differences observed in items related to an individual’s perceived control over situations. This suggests a growing sense of vulnerability and diminished capacity to manage emotions and concerns related to health—areas that should be targeted in psychological support interventions.

From a self-efficacy perspective, there was a notable decline in GSES scores, reflecting a diminished confidence in one’s ability to manage and resolve problems. The majority of participants reported decreased beliefs in their control over events, underscoring how chronic stress and uncertainty can erode personal coping resources. The significant reduction in self-efficacy, as measured by the GSES, aligns with findings from studies on chronic health risks [39], where prolonged stress and feelings of vulnerability impair individuals’ confidence in their coping abilities [40]. This decline in self-efficacy may foster a vicious cycle, leading to increased stress and perceived vulnerability.

The reduction in active and planning coping strategies from the Brief COPE further reflects a possible erosion of perceived operational control, consistent with the hypothesis that sustained stress undermines adaptive coping mechanisms. However, there was also a reduction in the use of dysfunctional strategies such as denial, which could indicate increased awareness of reality, although it cannot be excluded that this decline also reflects difficulty in mobilising effective coping resources over time.

Finally, perceived social support, measured through the MSPSS, showed a subtle but significant decrease—particularly in support from friends and other significant persons—while family support remained largely unchanged. This suggests that, over time, perceived support from one’s social network may diminish, potentially due to emotional fatigue or feelings of isolation. These aspects need to be addressed to maintain adequate social support, which is crucial in managing stress and health-related worries.

Based on the demographic and psychological characteristics of the 44 study participants, the study results highlight how, after a 36-month follow-up, the psychological and social features of the participants are closely linked to changes in perceived stress levels. These levels are categorised into three distinct clusters: Normal Stress, Stable Clinical Stress, and New Clinical Stress. This approach provides a better understanding of resilience and vulnerability factors within the context of genetic or familial risk for severe conditions such as PC.

The group with Normal Stress represents approximately one-third of the sample (32%) and is characterised by a high sense of self-efficacy, functional coping strategies, and a perceived positive social support, particularly from family. These factors appear to promote stable stress management over time, even in the presence of potentially traumatic life events, such as the loss of a family member, albeit on only one occasion. Their perception of competence, along with problem-solving and emotional regulation strategies, combined with a strong support network, is associated with a low incidence of dysfunctional coping and adequate psychological adaptability.

The group with Stable Clinical Stress (45%) exhibits signs of psychological distress, with moderate levels of self-efficacy and a more pronounced use of emotional coping strategies. Perceived social support is moderate, and the limited availability of support from friends may represent a vulnerability factor that could contribute to the persistence of high stress over time. This group highlights how the ability to manage stress can be influenced not only by personal resources but also by the quality of perceived social support, underscoring the importance of targeted interventions aimed at fostering stronger support networks.

The group with New Clinical Stress (22%) stands out for a profile characterised by high coping ability and very good perception of social support, as well as a moderate but stable level of self-efficacy. The emergence of new cases of high stress, despite good psychological and social functioning, suggests that external factors or unpredictable life events may act as triggers for the onset of clinically significant stress. The ability to adopt functional coping strategies and to maintain strong social support appears to serve as protective factors, although the appearance of new stress indicates the need for preventive interventions and ongoing monitoring.

An interesting aspect that emerged concerns the overall trend of decreasing self-efficacy and increasing feelings of helplessness over time, regardless of the perceived stress level. This dynamic suggests that, over the three years, even initially resilient individuals may experience a loss of perceived control and ability to cope with difficulties, highlighting the importance of continuous support strategies and interventions.

Following 3 years of surveillance for PC pancreatic cancer, 67% of the overall study population—comprising both the Stable Clinical Stress group and the New Clinical Stress group—exhibited high levels of perceived stress. Our findings reaffirm the critical role of psychological factors such as self-efficacy, coping strategies, and social support in shaping the evolution of perceived stress over time. It is essential to address not only the psychological burden but also to pay attention to psycho-oncological symptoms, with the aim of providing comprehensive and integrated care tailored to each individual.

In light of these findings, it is recommended that surveillance programmes for HRIs incorporate the following strategies:-implement annual psychological screenings to regularly evaluate the psychological burden, thereby facilitating early identification of needs and preventing the progression from adaptive to pathological stress responses;-promote targeted interventions for individuals experiencing distress—preferably group-based approaches such as relaxation techniques, mindfulness practices, cognitive-behavioural therapy focused on stress reduction, and resilience training—to support patients throughout the surveillance process, enhance their well-being, and improve stress management skills.

Fostering personal and social resources through these targeted interventions can help facilitate better stress management and reduce the risk of transitioning from adaptive coping to clinical psychopathology.

### 4.1. Broader Context and Future Research Directions

These findings underscore the intricate relationship between psychological distress, perceived control, and social support in shaping the well-being of HRIs.

An increase in feelings of helplessness coupled with decreased self-efficacy may create a feedback loop that amplifies stress levels and potentially adversely affects health behaviours, such as adherence to surveillance protocols. Furthermore, the reduction in social support can intensify feelings of isolation, highlighting the critical importance of maintaining strong support networks.

From a clinical perspective, these results emphasise the need for comprehensive psychosocial interventions tailored to the evolving needs of HRIs where the focus is on fostering an inevitable coexistence with uncertainty, which could be perceived not only as detrimental but also potentially transformative. In psychological terms, the shift in HRIs’ perspective lies in accepting what is beyond one’s control, reducing both avoidant coping and hyperfocus strategies, and investing personal energy in what can be changed and improved, as it falls within one’s sphere of mastery. Integrating stress management techniques, strategies to strengthen self-efficacy, and ongoing social support into routine care should be considered essential. For instance, counselling programmes that emphasise problem-solving skills and resilience-building may effectively help reduce the observed psychological declines.

Building on these insights, future research should investigate the effectiveness of targeted intervention programmes aimed at stabilising or reducing stress and increasing self-efficacy over time. Randomised controlled trials evaluating interventions such as cognitive-behavioural therapy, peer support groups, or digital health tools could identify best practices for psychosocial care in high-risk populations.

Additionally, longer follow-up periods and larger, more diverse samples are necessary to determine whether these psychological patterns persist, worsen, or improve following intervention. Qualitative studies exploring personal narratives could provide deeper insights into individual coping mechanisms and perceptions of social support, thereby informing more personalised approaches.

Finally, integrating biological markers of stress and immune function with psychosocial assessments may offer a more comprehensive understanding of how psychological well-being influences physical health outcomes in HRIs, paving the way for holistic, biopsychosocial intervention strategies.

### 4.2. Limits of the Study

In this study, several limitations related to the sample size and data collection methods have been identified, which must be carefully considered when interpreting the results. Firstly, the analysed sample is small, which restricts the ability to generalise the conclusions to a broader population. A limited sample size can also affect the statistical power of the study, reducing sensitivity in detecting significant differences or correlations between variables of interest.

Consequently, the findings should be interpreted with caution, taking into account the limited representativeness of the sample. Furthermore, the level of distress recorded among the subjects may be influenced by other critical factors or personal issues experienced by the participants, which were not fully controlled or isolated in the study. This overlap of stressors makes it difficult to attribute the observed distress solely to surgical surveillance, potentially introducing confounding factors into the results. Therefore, the data regarding the distress state might reflect, in addition to the specific variables investigated, other stress or discomfort conditions that the subjects were experiencing at the time of the interview.

Finally, the data collection method via telephone interviews presents an additional methodological limitation. Although this approach may have facilitated communication with individuals balancing work and family commitments, with limited available time, it also introduces the risk of bias. In particular, telephone communication can reduce the interviewer’s ability to perceive non-verbal cues or establish a trusting rapport with participants, elements that can influence the sincerity and accuracy of the responses provided. Issues related to comprehension or attention lapses on the part of the interviewees can compromise the quality of the data collected, leading to further distortions in the results. Furthermore, considering that 10 HRIs (8 out of 10 non-responders were under 65 years of age) did not answer the phone calls and were classified as lost to follow-up, it is reasonable to hypothesise that their unavailability for the interview was due to the interview’s length. In summary, although the study provides valuable and interesting insights, the limitations outlined above must be considered as factors influencing the validity and generalizability of the conclusions. Future research could benefit from a larger sample size, more comprehensive data collection methods, and more rigorous control of confounding variables to enhance the reliability and relevance of the emerging results

## 5. Conclusions

The findings from this longitudinal observational study underscore the importance of targeted psychological interventions aimed at helping HRIs to manage stress, strengthen self-efficacy resources, and sustain adequate social support. Future research should explore the mechanisms influencing these dynamics and evaluate the effectiveness of personalised psychosocial intervention programmes for individuals at high risk of serious illnesses like pancreatic cancer.

Counselling strategies, stress management education programmes, and engagement of support networks can serve as essential tools to improve the psychosocial well-being of this at-risk population in the long term.

In sum, this study highlights the importance of ongoing psychosocial support for high-risk individuals facing prolonged health threats. The observed trends suggest that without targeted interventions, psychological distress may intensify, potentially compromising both mental health and adherence to preventive strategies. Integrating psychological care into surveillance programmes is essential for promoting well-being and optimising health outcomes in this vulnerable group.

Future research should focus on intervention efficacy and personalised approaches to support resilience and adaptive functioning over time.

## Figures and Tables

**Figure 1 cancers-17-03014-f001:**
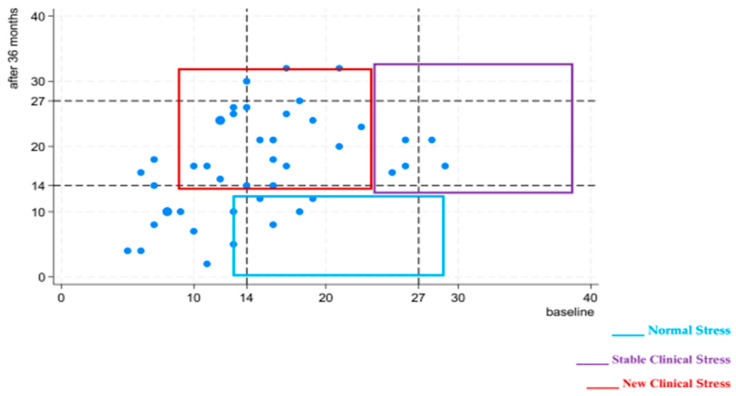
Frequency distribution of PSS10-scores at times of engagement and after 36 months (dotted lines indicate different levels of stress: 0–13 normal, 14–26 moderate and 27+ severe; marker size are weighed by frequencies).

**Figure 2 cancers-17-03014-f002:**
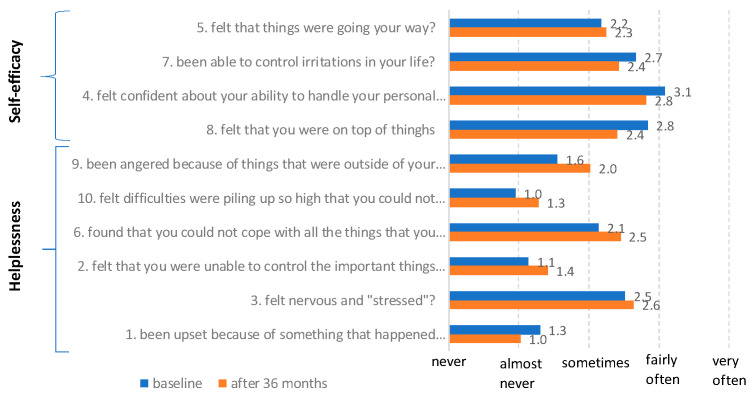
Mean values of the PSS-10 items at times of engagement and after 36 months, sorted by greatest difference within Perceived helplessness and lack of self-efficacy scales.

**Table 1 cancers-17-03014-t001:** Questionnaires and respective functioning areas.

Scale	Task	Number of Items	Item Range	Score Calculation	Score Range
PSS	Perceived Stress	10	0–4	sum of items	0–40
PSS-H	Perceived Helplessness	6	0–4	sum of items	0–24
PSS-L	Lack of Self-efficacy	4	0–4	sum of items	0–16
GSES	General Self-efficacy	10	1–4	sum of items	10–40
Brief-COPE	Coping strategies	28	1–4	sum of items	28–112
COPE-emotions	Emotion-focused strategies	10	1–4	sum of items	10–40
COPE-problems	Problem-focused strategies	6	1–4	sum of items	6–24
COPE-dysfunctional	Dysfunctional strategies	12	1–4	sum of items	12–48
MSPSS	Multidimensional Scale of Perceived Social Support	12	1–6	mean of items	1–6
MSPSS-family	Family support	4	1–6	mean of items	1–6
MSPSS-friends	Friend support	4	1–6	mean of items	1–6
MSPSS other-person	Significant person support	4	1–6	mean of items	1–6

**Table 2 cancers-17-03014-t002:** The observed differences in psychological distress: assessment after 36 months.

	Follow-Up	Baseline	Difference	St Dev	*t*-Test	*p*-Value
PSS-total	16.91	14.83	−2.08	7.63	−1.80	0.08
PSS- Perceived Helplessness	10.82	9.59	−1.23	5.79	−1.41	0.17
PSS- Lack of Self-Efficacy	6.09	5.24	−0.85	3.57	−1.57	0.12
GSES	28.09	31.95	3.86	4.44	5.77	**<0.01**
COPE-total	61.39	64.00	2.61	8.64	2.00	**0.05**
COPE-emotions	22.52	23.16	0.64	4.80	0.88	0.38
COPE-problems	17.45	18.80	1.34	3.28	2.71	**<0.01**
COPE-dysfunctional	21.41	22.05	0.64	4.80	0.88	0.38
MSPSS-total	4.27	4.63	0.36	1.16	2.04	**0.05**
MSPSS-family	4.67	4.76	0.08	1.51	0.35	0.73
MSPSS-friends	3.80	4.35	0.55	1.46	2.49	**0.02**
MSPSS-other persons	4.33	4.79	0.45	1.52	1.95	0.06

Statistically significant differences between baseline and follow-up are in bold.

**Table 3 cancers-17-03014-t003:** The Brief COPE coping strategies: a description of the changes after a 36-month period.

STRATEGIES	Follow-Up	Baseline	Difference	St Deviation	*t*-Test	*p*-Value
EMOTIONAL-FOCUSED						
Use of emotional support	4.14	4.33	0.18	2.0	0.60	0.55
Positive reframe	4.52	4.89	0.36	2.0	1.20	0.23
Acceptance	6.30	6.45	0.16	1.5	0.71	0.48
Religion	3.64	3.91	0.27	1.8	0.99	0.32
Humour	3.93	3.57	−0.36	1.6	−1.49	0.14
PROBLEM-FOCUSED						
Active coping	6.55	7.07	0.52	1.8	1.88	0.07
Planning	6.50	7.18	0.68	1.4	3.28	**<0.01**
Use of instrumental support	4.41	4.48	0.07	1.8	0.26	0.80
DYSFUNCTIONAL COPING						
Venting	4.59	4.68	0.09	1.9	0.28	0.78
Denial	2.32	2.84	0.52	1.5	2.38	**0.02**
Substance use	2.0	2.0	0.00		n.c.	
Behavioural disengagement	2.43	2.50	0.70	1.4	0.33	0.74
Self-distraction	4.70	5.00	0.30	2.5	0.77	0.45
Self-blame	5.36	5.02	−0.34	1.8	−1.24	0.22

Bold = significant difference (baseline vs. follow-up). n.c. = not calculable.

**Table 4 cancers-17-03014-t004:** Participants’ demographic and psychological characteristics categorised by stress levels, based on a PSS cutoff score of 14.

	Normal Stress (PSS fu < 14)	Stable Clinical Stress (PSS fu > 14 & PSS bs > 14)	New Clinical Stress (PSS fu > 14 & PSS bs < 14)
N (%)	14 (32%)	20 (46%)	10 (22%)
PSS baseline score	11.3 (4.5)	19.6 (4.9)	10.3 (2.7)
Age	54.6 (10.5)	56.3 (11.7)	59.9 (9.4)
Gender: female male	9 (64%) 5 (36%)	12 (60%) 8 (40%)	8 (80%) 2 (20%)
Family member Loss: yes	1 (7%)	6 (30%)	0 (00%)
GSES FU score	32.2 (5.8)	25.8 (5.4)	27.0 (3.4)
COPE total FU score	61.1 (8.8)	58.4 (10.0)	67.7 (7.2)
COPE-emotions FU score	23.2 (5.1)	20.7 (6.3)	25.3 (5.7)
COPE-problems follow-up score	17.6 (2.9)	16.4 (3.4)	19.4 (1.5)
COPE-dysfunctional follow-up score	20.4 (2.8)	21.4 (3.2)	23.0 (2.4)
MSPSS total FU score	4.7 (0.9)	3.6 (1.3)	5.0 (0.4)
MSPSS family FU score	5.3 (1.0)	3.9 (1.4)	5.5 (0.5)
MSPSS friends FU score	4.1 (1.0)	3.3 (1.3)	4.4 (0.7)
MSPSS other-person FU score	4.8 (1.0)	3.6 (1.6)	5.2 (0.4)

Categorical variables are expressed by frequency and percentage, continuous ones by mean and std deviation.

## Data Availability

The data that support the findings of this study are available on request from the corresponding author. The data are not publicly available due to privacy or ethical restrictions.

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
