# Peer review of "Assessment of Psychological Burden in Individuals with Hereditary Risk of Pancreatic Cancer Under Surveillance: Evaluation of Distress 3 Years After Enrollment"

_cancers, 2025, doi:10.3390/cancers17183014_

Round 1
Reviewer 1 Report
Comments and Suggestions for Authors
The manuscript offers a valuable and timely contribution to the literature on the psychological burden experienced by individuals with hereditary risk of pancreatic cancer who undergo long-term surveillance. The longitudinal design, combined with the use of validated instruments, provides important insights into how stress, self-efficacy, coping strategies and social support evolve over time in this vulnerable population.
The text is overall clear, but the English language would benefit from further refinement. In particular, there are occasional inconsistencies in grammar, punctuation, and formatting (for example, spacing around symbols, alignment of subheadings, and article use). A professional proofread or careful copyediting would enhance readability and presentation. Similarly, figures and tables are generally informative, but the captions could be more descriptive so that each element can stand independently, and minor adjustments in formatting (such as consistent use of decimals and fonts) would improve clarity.
The methods are adequately described, but the section could be made clearer by providing more detail on the telephone interview procedures, including the typical length of interviews, the training of interviewers, and the level of standardization across participants. Given that the telephone format is acknowledged as a limitation, elaborating on these aspects would help readers better assess the reliability of the data collected.
Finally, the discussion is mostly comprehensive, but it would be strengthened by explicitly linking the identified stress clusters (Normal Stress, Stable Clinical Stress, and New Clinical Stress) with potential intervention strategies. Providing concrete suggestions on how different psychosocial care pathways could address the specific needs of these groups would increase the translational impact of the findings.
Overall, the study is well designed, scientifically sound and of high relevance to both psycho-oncology and surveillance programs in high-risk populations. The revisions needed are minor, focusing primarily on language polishing and small clarifications that would make the manuscript even more accessible and impactful.
Author Response
Please see the attachmen

Reviewer 2 Report
Comments and Suggestions for Authors
This is a very interesting and well implemented study. The fact that almost 2/3 of the study participants continue to experience high levels of stress and anxiety is interesting. Combined with the fact that they experience more helplessness and less confidence in their ability to solve problems, this highlights the need for psychological intervention to be available to this group. In addition, it would be interesting in follow up studies to examine the coping mechanisms and strengths of the 1/3 of the study group that did not experience clinical distress. This might help in designing future interventions focusing on helping this population of high risk individuals. The only concern I have is the high (almost 20%) lack of response rate. There is also one sentence where the authors note something related to "loss of a family member due to CP." (I'm assuming they meant PC, but this needs to be changed.
Author Response
Please see the attachmen
